# The role of citizen science in mosquito-borne disease surveillance and control: A scoping review

Nima Kianfar[1], Kimia Savoji[2], Xiao Huang[3], Di Yang[4], Abe Mollalo[5]*

1 Department of Geospatial Information Systems, Faculty of Geomatics Engineering, K. N. Toosi University of Technology, Tehran, Iran, 2 School of Computing, Clemson University, Clemson, South Carolina, United States of America, 3 Department of Environmental Sciences, Emory University, Atlanta, Georgia, United States of America, 4 Department of Geography, University of Florida, Gainesville, Florida, United States of America, 5 Department of Public Health Sciences, Medical University of South Carolina, Charleston, South Carolina, United States of America

* mollalo@musc.edu

## Abstract

### Background

Mosquito-borne diseases (MBDs) pose substantial global health and economic burdens. Although conventional MBDs surveillance systems remain essential, they are often resource-intensive, uneven in coverage, and often insufficiently responsive to spatio-temporal variations in mosquito presence and risk. Citizen science, increasingly enabled by mobile and digital technologies, offers a scalable complement to expand surveillance reach and timeliness. However, existing reviews have not comprehensively integrated evidence across diverse dimensions of citizen science applied to MBDs surveillance and control.

### Methods

We conducted a systematic search of PubMed, MEDLINE, CINAHL, Scopus, and Web of Science from January 1, 2000, to October 17, 2025, to identify peer-reviewed studies examining citizen science applications in MBDs surveillance and control. Data were extracted and synthesized on study characteristics, participation objectives, recruitment strategies, citizen-generated data and technologies, validation mechanisms, effort-bias handling, analytical approaches, public health outputs, reported biases and methodological limitations, and ethical and governance practices.

### Results

Of 3,734 records identified, 61 studies met inclusion criteria, with most published after 2017 (93.4%). Studies were conducted in Europe (44.3%) and the Americas (21.3%), with minimal representation from Asia (3.3%). Malaria-related surveillance was most common (23.0%), followed by dengue (13.1%), with other mosquito-borne

**Data availability statement:** All relevant data are within the manuscript and its Supporting information files (S2 File).

**Funding:** A.M. is supported by the National Aeronautics and Space Administration (NASA) under award number 80NSSC25K7190.

**Competing interests:** The authors have declared that no competing interests exist.

diseases examined only sporadically, including West Nile virus (4.9%), Usutu virus (1.6%), La Crosse virus (1.6%), and California serogroup viruses (1.6%). Most studies were conducted in urban settings (47.5%), followed by mixed urban–rural contexts (36.1%), with relatively few exclusively in rural areas (18.0%). Mosquito Alert was the most frequently reported platform (23.0%), followed by GLOBE Observer (13.1%) and iNaturalist (11.5%). Commonly reported outputs included trend analyses (52.5%), risk-factor identification (44.3%), spatial predictions (42.6%), hotspot mapping (19.7%), and risk modeling (16.4%). Reporting of ethical and governance practices was inconsistent across studies.

## Conclusions

The growing body of evidence indicates that citizen science can enhance mosquito surveillance, particularly for monitoring invasive species and spatio-temporal trends. Nevertheless, gaps in methodological rigor, representativeness, and ethical transparency limit its broader operational use.

---

## 1.  Introduction

Mosquito-borne diseases (MBDs) remain major global public health threats, imposing substantial and continuing health burdens [1]. Each year, these infections account for over 347 million cases and more than 447 thousand deaths worldwide [2,3]. Beyond their health impacts, the associated economic burden is considerable across diverse settings. A recent global synthesis estimated that invasive *Aedes* mosquitoes and the diseases they transmit resulted in a cumulative cost of at least USD 94.7 billion between 1975 and 2020. Notably, the study documented a 14-fold increase in reported costs over time, with average annual costs of USD 3.1 billion, while emphasizing substantial underreporting and underestimation [4].

Effective prevention and control of MBDs depend on surveillance systems capable of detecting changes in vector presence and transmission risk. Despite the magnitude and growing burden of MBDs, traditional mosquito surveillance remains uneven in coverage, resource-intensive, and often unable to resolve fine-scale or rapidly changing spatial patterns [5,6]. Addressing these limitations requires surveillance systems that are timely, spatially sensitive, and capable of tracking changes in vector presence, abundance, and spread. In response to these challenges, citizen science, enabled by digital tools and mobile technologies, offers a practical complement to traditional entomological and public health surveillance [7]. By engaging large numbers of participants, citizen science can expand the geographic and temporal reach of monitoring, particularly through smartphone-based reporting, geotagged observations, and image-based submissions [8]. Several well-established platforms illustrate the feasibility of this approach. For example, Mosquito Alert enables the public to submit geolocated mosquito photographs that can be validated by experts and used for monitoring invasive and medically important species [9]. Similarly, the GLOBE Observer Mosquito Habitat Mapper supports structured reporting of larval habitats

and standing-water sites through a smartphone-based interface, generating standardized data streams that can inform surveillance and mitigation activities [10].

Citizen science is particularly well-suited to MBDs surveillance as many actionable indicators of mosquito risk are observable in everyday environments, such as household and neighborhood breeding sites, nuisance hotspots, and encounters with invasive species [11]. Community reporting and engagement can strengthen early detection and broaden coverage in areas where routine surveillance is intermittent or resource-constrained, while also supporting sustained monitoring across seasons and years [12]. At the same time, advances in quality assurance and validation workflows, including automated classification supported by expert review, have improved the feasibility of translating citizen-submitted mosquito data into surveillance-relevant information [2].

Despite this growing body of literature, the intersection of citizen science and MBDs surveillance remains an incompletely synthesized area of research. Although prior reviews have examined selected aspects of citizen science-based mosquito surveillance [13,14], they do not provide an integrated overview capturing participation models, citizen-generated data types, platforms and tools, analytical approaches, epidemiologic outputs, and ethical and operational considerations across MBDs surveillance and control studies. To address this gap, we conducted this scoping review to synthesize peer-reviewed evidence on citizen science applications in MBDs surveillance and control. Specifically, we aimed to systematically:

1. Describe citizen science participation and implementation activities;

2. Summarize recruitment and participation objectives (e.g., surveillance, education, vector control);

3. Characterize data and enabling technologies (disease/vector focus, platforms/tools, and citizen-generated data types); and

4. Synthesize analytical approaches, validation procedures, participation biases and mitigation strategies, outputs, and ethical considerations.

Together, this review will identify methodological patterns, recurrent challenges, and critical gaps that can inform future study design and the integration of citizen-generated data into MBDs surveillance and control, including opportunities to strengthen early warning systems and predictive modeling.

## 2. Methods

This scoping review adhered to the methodological framework outlined by the Preferred Reporting Items for Systematic Reviews and Meta-Analyses Extension for Scoping Reviews (PRISMA-ScR) [15]. The completed PRISMA checklist is available in the S1 File.

### 2.1. Search strategy

A comprehensive literature search was conducted across several databases: PubMed, MEDLINE, CINAHL, Scopus, and Web of Science. The search covered studies published from January 1, 2000, to October 17, 2025, to identify English-language, peer-reviewed studies on citizen science applications in MBDs surveillance and control. The search period beginning in 2000 was selected to align with the emergence of digital and participatory surveillance approaches driven by expanding internet and mobile access. Our search strategy combined controlled vocabulary and free-text keywords from two primary conceptual domains: citizen science and MBDs. Broad citizen science-related terms were intentionally included to maximize sensitivity and capture the diverse terminology used to describe citizen-involved surveillance across disciplines. No restrictions were applied on geographic settings or population characteristics. The full search key terms for all databases are provided in Table 1 that were adopted and refined from those used in prior reviews [13,16].

**Table 1. Key terms used across databases.**

| Theme | Key Terms |
| --- | --- |
| Citizen Science | ("citizen science" OR "participatory science" OR "community science" OR "community engagement" OR "community participation" OR "public participation" OR "community-based monitoring" OR "participatory monitoring" OR "volunteer monitoring" OR "participatory surveillance" OR "crowdsourcing" OR "mobile crowdsensing" OR "volunteered geographic information" OR "VGI") |
| AND | |
| Mosquito | ("mosquito" OR "mosquito surveillance" OR "mosquito monitoring" OR "mosquito borne disease" OR "vector" OR "vector borne disease" OR "vector surveillance" OR "entomological surveillance" OR "vector control" OR "mosquito control" OR "Aedes" OR "Anopheles" OR "Culex" OR "dengue" OR "malaria" OR "Zika" OR "chikungunya" OR "West Nile" OR "yellow fever" OR "arbovirus") |

## 2.2. Eligibility criteria

In this review, citizen science in MBDs was defined as the active participation of non-professional individuals in generating, reporting, or interpreting mosquito-related data that are used to inform surveillance or vector-control activities through the scientific process [17]. Based on this definition, studies were included if they met all the following criteria: i) incorporated citizen science or participatory components; ii) focused on MBDs; iii) examined surveillance or vector-control as a primary study objective; iv) were original peer-reviewed studies; and v) were published in English. Studies were excluded if they i) were not peer-reviewed (e.g., conference abstracts, dissertations, reports, grey literature, reviews, and commentaries); ii) were unrelated to MBDs; and iii) were assessed mosquito-related data without involving active citizen participation.

## 2.3. Screening process and study selection

All retrieved studies were imported into Covidence (https://www.covidence.org/) for reference management. The screening process was conducted in two stages. In the first stage, titles and abstracts were independently reviewed by two reviewers (NK and KS) to minimize selection bias and exclude the studies that lacked a citizen science component or did not address MBDs surveillance or control. In the second stage, the full text of selected studies was thoroughly assessed against the predefined eligibility criteria. Any discrepancies were resolved through discussion during regular meetings and, when necessary, adjudication by a senior reviewer (AM).

## 2.4. Data extraction

Following study selection, a standardized data extraction form was developed using Microsoft Excel (Microsoft Corporation, Redmond, WA, United States) to systematically capture information aligned with the review objectives. Extracted data were collected on: i) study characteristics (bibliographic information, geographic location, study setting, and study design); ii) the citizen science framework (participant sample size, demographic characteristics, level of citizen involvement, recruitment methods, and purpose of participation); iii) MBDs focus, validation mechanisms, and analytical approaches (disease(s) studied, data types collected, validation procedures, effort-bias handling, analytical or modeling methods, and epidemiologic outputs); iv) platforms and technologies used (mobile applications, web portals, GPS-enabled tools, smartphones, sensors); and v) reported biases, methodological limitations, and ethical considerations (e.g., participation and sampling biases, data quality issues, validation limitations, recruitment/engagement challenges, representativeness limitations, technology-related barriers, spatial bias, ethical procedures, and data availability). Data extraction was independently completed by two reviewers (NK and KS), and any disagreements were resolved by consensus. The full extraction table is provided in the S2 File.

## 2.5. Data synthesis

Extracted data were synthesized using a structured, descriptive approach aligned with the review objectives. The selected studies were categorized based on study characteristics, MBDs focus, citizen science approach, platforms and technologies used, validation mechanisms, effort-bias handling, analytical approaches, reported biases and methodological limitations, and ethical considerations. This structured synthesis enabled the identification of methodological patterns, recurrent challenges, and key gaps in the implementation of citizen science approaches for MBDs surveillance and control.

## 2.6. Ethics statement

Because this study is a scoping review of published literature and did not involve human participants, identifiable personal data, or primary data collection, ethics approval and informed consent were not required.

## 3. Results

### 3.1. Study selection

Our search initially identified 3,734 studies across the databases. A total of 2,155 duplicates were automatically removed by Covidence or manually, leaving 1,579 studies for title and abstract screening. At this stage, 1,309 studies were excluded, primarily because they did not involve citizen science participation, were not related to MBDs or did not address surveillance and control. The remaining 270 studies underwent full-text assessment, after which 209 studies were excluded for the following reasons: absence of a citizen science component (n = 89), lack of focus on MBDs (n = 47), primary outcome not related to surveillance or control (n = 40), non-original articles (n = 26), and non-English full text (n = 7). In total, 61 studies met all eligibility criteria and were included in this scoping review. Fig 1 presents the PRISMA flow diagram showing the study selection process.

### 3.2. Spatial and temporal distribution

The included studies were geographically diverse, spanning all continents. Europe accounted for the largest proportion of studies (n = 27, 44.3%), spanning 9 countries, including Spain [2,7,18–25], Germany [26–31], Italy [9,32,33], Hungary [34,35], France [12], Austria [36], the Netherlands [37], Ireland and the United Kingdom [38]. The Americas contributed 13 studies (n = 13, 21.3%), across six countries, including the United States [10,11,39–44], Canada [45], Cuba [46], Haiti [47], Nicaragua and Mexico [48,49]. Africa contributed 12 studies (n = 12, 19.7%), spanning seven countries and regions, including Rwanda [50–52], Tanzania [53–55], Malawi [56], Ethiopia [57], the Gambia [58], Madagascar [41], and West Africa-focused [59], and Africa-wide multi-country initiatives [60]. Oceania contributed seven studies (n = 7, 11.5%), representing Australia [61–65] and the Solomon Islands [66,67]. Asia was represented by two studies (n = 2, 3.3%), both in Sri Lanka [68,69].

In addition to region-specific studies, two studies (n = 2, 3.3%) were classified as global in scope, drawing on data from worldwide surveillance platforms [70,71], and six studies (n = 6, 9.8%) were categorized as multinational, as they involved more than one country [2,41,45,59,60,72]. Overall, the included studies covered 24 unique countries. The spatial distribution of studies is depicted in Fig 2A.

Temporally, few studies were published before 2017 (n = 4, 6.6%) [46,48,53,68], whereas the majority were published from 2017 onward (n = 57, 93.4%). A notable rise occurred after 2020 (n = 42, 68.9%). Fig 2B illustrates the temporal distribution of included studies, with publications peaking in 2021 and remaining elevated in subsequent years.

### 3.3. Mosquito surveillance

Among the included studies, mosquito monitoring was the most common focus (n = 40, 65.6%), primarily concentrated on mapping the distribution, abundance, and spread of mosquito taxa, including species described by the original studies as invasive or nuisance in specific geographic and surveillance contexts. *Aedes albopictus* was the most commonly reported

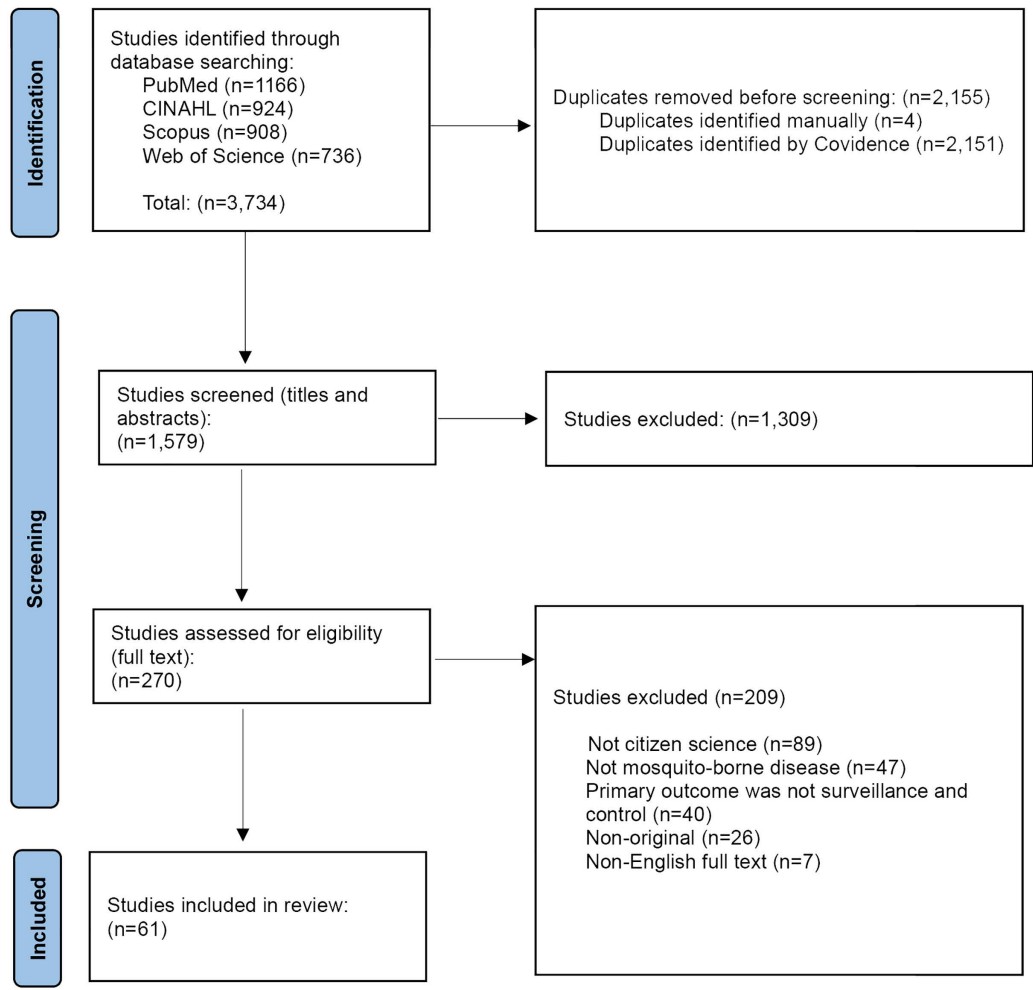

**Fig 1. PRISMA flow diagram summarizing the study selection process.**

species (n = 16, 26.2%) [2,7,9,12,18,20,22–24,31–36,41], followed by *Aedes japonicus* (n = 6, 9.8%) [2,19,21,22,34,35], *Aedes koreicus* (n = 5, 8.2%) [2,9,28,34,35], and *Aedes aegypti* (n = 3, 4.9%) [2,22,43]. Additional species included *Aedes notoscriptus* (n = 1, 1.6%) [62] and *Culex* species, most commonly *Culex pipiens* (n = 4, 6.6%) [2,9,31,37] and *Culex quinquefasciatus* (n = 1, 1.6%) [62].

A smaller subset of studies focused on *Anopheles* species, including *Anopheles plumbeus* (n = 2, 3.3%) [26,27], members of the *Anopheles gambiae* complex (*A. gambiae* s.s., *A. arabiensis*, *A. funestus;* n = 1, 1.6%) [56], and mixed *Anopheles* larvae observations through community surveillance platforms (n = 3, 4.9%) [10,70,71].

Malaria-related surveillance was reported in 14 studies (n = 14, 23.0%), nearly all of which focused on dominant *Anopheles* vectors [25,47,50–60,68]. Dengue-related surveillance appeared in eight studies (n = 8, 13.1%) [2,46,48,49,59,66,67,69], while other MBDs were documented less frequently, including West Nile virus (n = 3, 4.9%) [21,31,37], Usutu virus (n = 1, 1.6%) [37], La Crosse virus (n = 1, 1.6%) [40], and California serogroup viruses (n = 1, 1.6%) [45].

Regional patterns were evident. Most African studies focused on malaria vectors (n = 10 of 12, 83.3%), whereas European studies primarily emphasized *Aedes* species and general mosquito surveillance (n = 17 of 27, 63.0%). Several

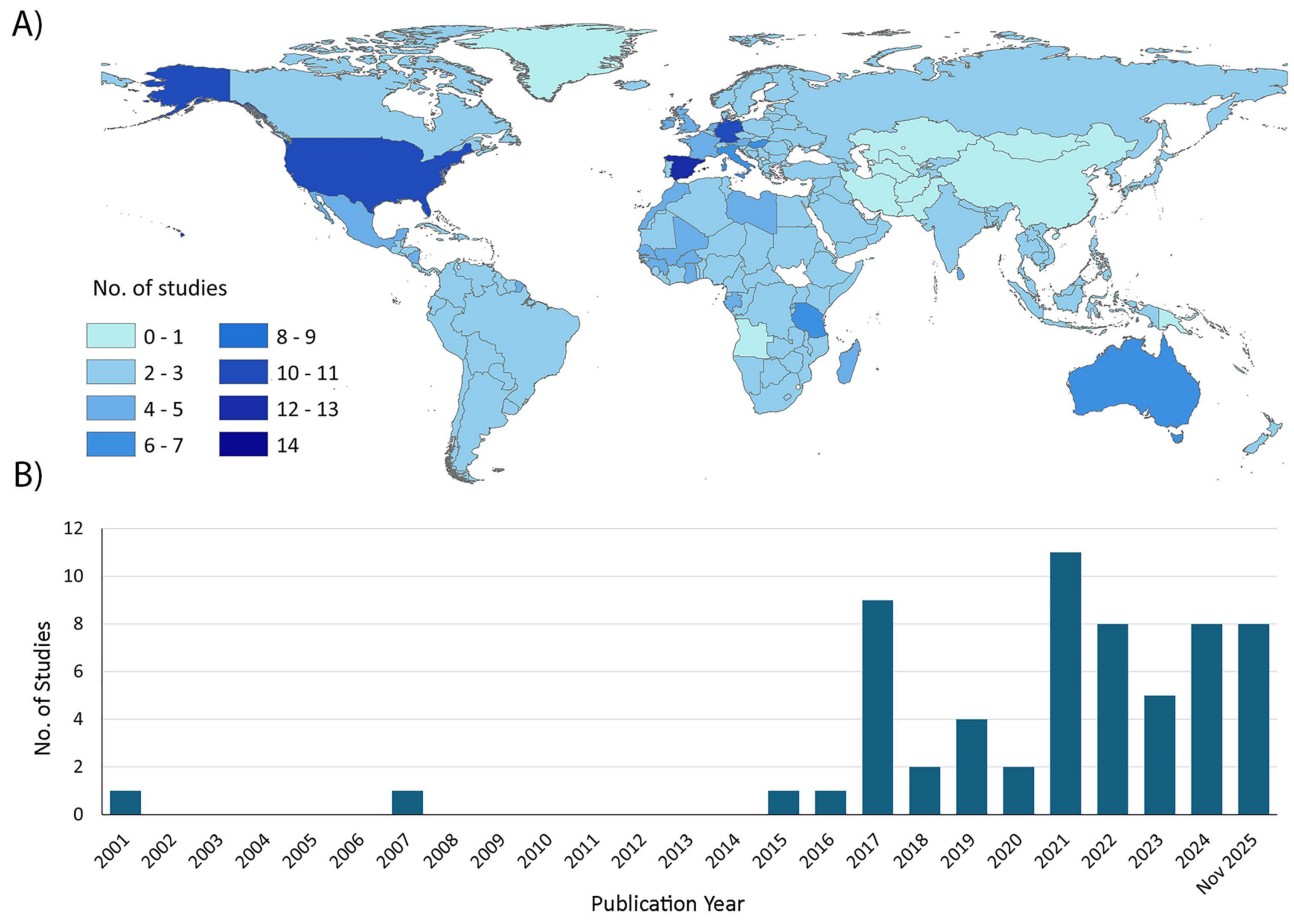

**Fig 2. Spatial (A) and temporal (B) distribution of the included studies on citizen science–based MBDs surveillance and control.** Panel A shows all countries represented across the included studies (n = 24 unique countries), including those captured through multi-country study designs. The base map shapefiles used in Panel A were derived from publicly available administrative boundary data obtained from Natural Earth (https://www.naturalearth-data.com/), which is in the public domain.

studies from the Americas (n = 5 of 13, 38.5%) and Oceania (n = 2 of 7, 28.6%) also focused on dengue, Zika, or chikungunya surveillance.

### 3.4. Study designs and level of analysis

Study designs varied in how citizen science was used, ranging from descriptive surveillance to system evaluation and, less commonly, intervention and feasibility studies. Descriptive study designs were the most common, appearing in 46 studies (n = 46, 75.4%), primarily focused on mosquito activity and spatio-temporal patterns in citizen-submitted data. Evaluation-oriented studies appeared in 15 studies (n = 15, 24.6%), most commonly assessing the performance, accuracy or reliability of citizen science surveillance systems. Intervention-based designs were identified in 10 studies (n = 10, 16.4%), including three randomized controlled trials (n = 3, 4.9%) [12,48,49]. Feasibility studies were identified in six studies (n = 6, 9.8%) [40,49,50,61,66,67], and four studies described pilot implementations (n = 4, 6.6%) [41,47,66,67].

Most studies were conducted in urban settings (n = 29, 47.5%), while fewer were implemented across mixed urban-rural contexts (n = 22, 36.1%). A smaller group of studies was carried out exclusively in rural environments (n = 11, 18.0%).

Sample sizes varied substantially across studies. Very small studies (≤50 participants) were identified in three studies (n = 3, 4.9%), while small-to-medium studies (51–500 participants) were identified in 15 studies (n = 15, 24.6%). Medium-to-large studies (501–10,000 participants) were reported in 14 studies (n = 14, 23.0%), and large-scale studies involving more than 10,000 users or submissions were documented in eight studies (n = 8, 13.1%). Sample size was not reported in 21 studies (n = 21, 34.4%).

Participants' gender distribution was reported in only eight studies (n=8, 13.1%). Among these, several studies reported approximately balanced male and female participation (n=5, 8.2%) [52–54,56,61], while others noted higher levels of female participation (n=3, 4.9%) [44,50,63].

### 3.5. Citizen science involvement

Citizen involvement varied both in intensity and the degree to which it contributed to different surveillance functions, ranging from simple reporting of observations to more active roles in local monitoring, intervention, annotation, and basic analytic support.

**3.5.1. Data collection.** Citizen participation was predominantly focused on data collection (n = 58, 95.1%). Across these studies, citizens contributed a range of data types. Presence/absence data, commonly used for mapping species distributions or modeling occurrence patterns, were reported in 26 studies (n = 26, 42.6%). Photographic submissions were reported in 12 studies (n = 12, 19.7%). Habitat or breeding-site documentation, including reports of larval habitats, breeding sites, or nuisance locations, was reported in 11 studies (n = 11, 18.0%). Trap-based data collection was also reported in 11 studies (n = 11, 18.0%), most commonly using ovitraps (n = 5 of 11, 45.5%). Other trap types included BG-Sentinel traps (n = 3, 27.3%), BG Gravid *Aedes* Traps (BG-GAT) (n = 3, 27.3%), $CO_2$-baited traps (n = 2, 18.2%), and BG-Mosquitaire and CDC light traps (n = 1 each, 9.1%). Some studies used multiple trap types, so the total number of trap occurrences exceeds the number of studies.

Submission of physical mosquito specimens or eggs by citizen participants was also reported in 10 studies (n = 10, 16.4%). Acoustic recordings of mosquito flight tones were reported in only three studies (n = 3, 4.9%). In general, simpler citizen-contributed data types such as presence reports and photographs were commonly used to support occurrence mapping, species identification, and early detection. In contrast, trap-based, specimen-based, and acoustic data were more often used in structured monitoring or more specialized validation and analytic workflows.

**3.5.2. Research engagement and participation.** Beyond data collection, citizen engagement in research-related activities was reported in 22 studies (n = 22, 36.1%) (Fig 3A). These activities included participation in community-based planning or coordination with local leaders or organizations (n = 10, 16.4%), involvement in workshops and training sessions (n = 8, 13.1%), collaboration with research teams or project staff (n = 8, 13.1%), and contributions of experiential or local ecological knowledge (n = 5, 8.2%).

**3.5.3. Monitoring and intervention.** Active participation in monitoring or intervention activities, such as larval source management, trap maintenance, habitat reduction, or follow-up surveillance, was reported in 18 studies (n = 18, 29.5%). These activities included community-led larviciding (n = 6, 9.8%) [12,47–49,56,58], breeding-site elimination or habitat reduction (n = 6, 9.8%) [10,50,57,60,66,70], and maintenance of trapping or monitoring infrastructure (n = 5, 8.2%) [22,31,40,46,69]. Compared with observational reporting platforms, these studies positioned citizens as active co-implementers of surveillance and control, linking participation more directly to local vector management and public-health action.

**3.5.4. Dataset annotation, validation, and analysis.** Dataset annotation was documented in eight studies (n = 8, 13.1%) [7,11,22,35,39,60,61,64], in which citizen-contributed data (often photographic or app-based observations) were reviewed, labeled, or categorized to support species identification, model training, or quality control workflows. These annotation activities functioned as Human-in-the-Loop quality assurance steps, with citizen contributions supporting validation and downstream analytical workflows. Most annotation activities supported species identification workflows

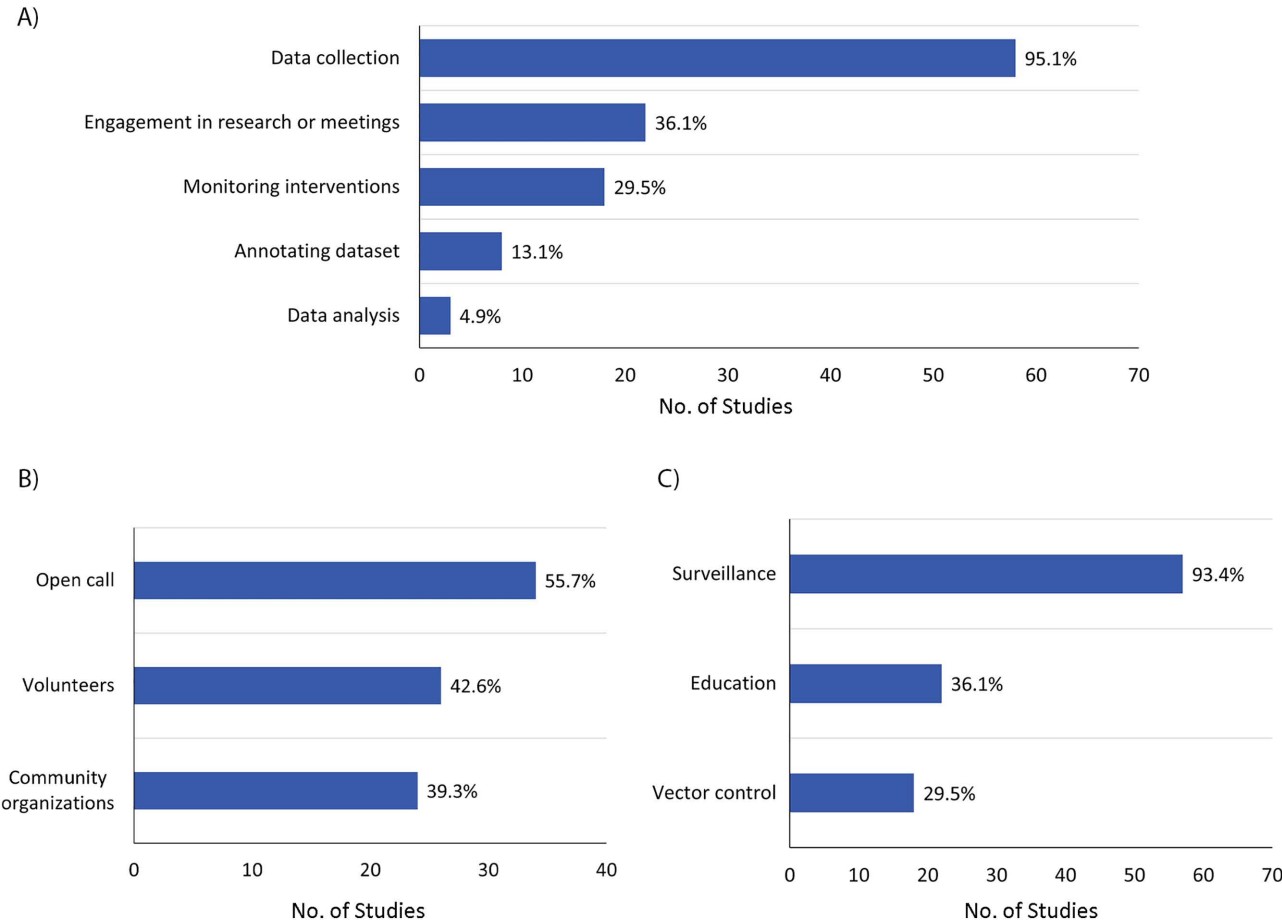

**Fig 3. Characteristics of the citizen science–based MBDs among the included studies: (A) level of involvement, (B) recruitment strategy, and (C) participation purpose.**

(n = 6, 9.8%) [7,11,22,60,61,64], while a smaller number contributed to model development or dataset structuring (n = 2, 3.3%) [35,39]. A limited subset of studies (n = 3, 4.9%) engaged citizens directly in data analysis [23,39,40], including analytic interpretation, classroom-based data exercises, or community-supported modeling activities integrated into educational programs or surveillance evaluations.

**3.5.5. Recruitment strategies.** Recruitment strategies varied across studies (Fig 3B). Open-call recruitment was the most common approach (n = 34, 55.7%), most often conducted through mobile applications (n = 15, 24.6%) [2,7,9,11,19,21–24,31,33,59,64,65,71], national or local media (n = 7, 11.5%) [9,10,31,33,34,37,43], public websites or email invitations (n = 5, 8.2%) [26,27,30,35,44], social media campaigns (n = 4, 6.6%) [10,45,61,63], and institutional newsletters or academic channels (n = 3, 4.9%) [44,61,63].

Volunteer-based recruitment was reported in 26 studies (n = 26, 42.6%) and involved self-selected participants motivated by local interest or concern (n = 10, 16.4%) [18,20,25,28,41,47,52,62,67,72], prior engagement with community programs (n = 9, 14.8%) [32,40,48–50,53,58,60,66], or outreach through workshops, campaigns, or field activities (n = 6, 9.8%) [20,40,50,52,58,70].

Recruitment through community organizations was documented in 24 studies (n = 24, 39.3%), including engagement with municipalities or local government units (n = 6, 9.8%) [32,40,46,49,51,58], village leadership (n = 6, 9.8%)

[53–56,58,69], local health authorities (n = 4, 6.6%) [46,51,57,58], schools or educational institutions (n = 4, 6.6%) [10,39,40,45], and community health workers (n = 2, 3.3%) [51,58]. Academic institutions supported recruitment in two studies (n = 2, 3.3%) [61,69]. In general, open-call recruitment was most common in app- and platform-based surveillance systems designed to maximize submission volume and geographic reach. In contrast, recruitment through community organizations, schools, and local leadership was more characteristic of place-based initiatives that depended on sustained engagement, training, or intervention activities.

Recruitment strategies also differed in how directly studies addressed structural participation imbalance across geographic contexts. In 33 studies (n = 33, 54.1%), recruitment, engagement, or outreach approaches clearly or partly aimed to broaden participation beyond highly populated settings. Of these, seven studies (n = 7, 11.5%) implemented more explicit targeted strategies for rural, remote, or underserved areas [32,42,47–50,57]. Within these seven studies, common approaches included partnerships with schools or school-based networks (n = 6, 9.8%), engagement with community leaders, local authorities, or community health workers (n = 6, 9.8%), door-to-door or household visits (n = 3, 4.9%), village-based recruitment structures (n = 2, 3.3%), and low-access tailored measures such as shared smartphones, offline data collection, or local-language outreach (n = 1, 1.6%) (non-mutually exclusive). The other 26 studies (n = 26, 42.6%) addressed this issue more indirectly through locally adapted outreach, media or social media campaigns, expanded recruitment networks, or discussions of ways to strengthen participation in low-contribution regions, without clearly framing these efforts as explicit corrections for urban or population-density bias. In the remaining 28 studies (n = 28, 45.9%), no clear recruitment-side strategy to reduce structural participation imbalance was reported. These patterns indicate that the recruitment strategy was not merely an operational detail, but a component of system design that shaped who participated, where data were generated, and how vulnerable the resulting outputs were to structural participation imbalance.

**3.5.6. Purpose of citizen participation.** Surveillance activities were the predominant purpose of citizen participation (n = 57, 93.4%) (Fig 3C). These contributions included species identification (n = 55, 90.2%), presence/absence reporting (n = 41, 67.2%), inputs supporting risk assessment and spatial modeling (n = 17, 27.9%), trap monitoring (n = 12, 19.7%), habitat or breeding-site documentation (n = 11, 18.0%), early detection of invasive species (n = 8, 13.1%), nuisance documentation (n = 6, 9.8%), and acoustic detection (n = 3, 4.9%).

Educational components were reported in 22 studies (n = 22, 36.1%), including public training sessions (n = 10, 16.4%) [32,33,39,40,46,50,56,63,69,70], community workshops focused on vector biology and prevention practices (n = 7, 11.5%) [32,39,40,51,55,56,63], school-based learning activities (n = 7, 11.5%) [10,39,40,44,56,66,70], awareness campaigns (n = 6, 9.8%) [32,33,40,69–71], and classroom-based mosquito ecology instruction (n = 5, 8.2%) [39,40,44,56,66].

Vector control objectives were present in 18 studies (n = 18, 29.5%), with citizen involvement in targeted interventions to reduce mosquito populations or limit disease transmission. These include direct intervention activities (n = 10, 16.4%) [11,12,18,22,32,35,40,46,56,66], larval source management (n = 7, 11.5%) [47–49,51,58,66,70], breeding-site elimination (n = 7, 11.5%) [22,32,40,47,56,58,70], trap deployment or maintenance (n = 6, 9.8%) [12,36,40,48,49,66], and community cleanup activities (n = 4, 6.6%) [40,46,56,70]. Overall, citizen science initiatives commonly combined multiple functions, with some platforms more strongly oriented toward quantitative surveillance and others placing greater emphasis on education, engagement, or local control while still contributing surveillance-relevant information.

## 3.6. Data types, tools and platforms

**3.6.1. Data types.** Image-based submissions were the most common form of citizen-generated data (n = 33, 54.1%). These images were used for species identification (n = 19, 31.1%), presence documentation (n = 14, 23.0%), habitat characterization (n = 11, 18.0%), and nuisance assessments (n = 4, 6.6%). Larval counts were reported in 22 studies (n = 22, 36.1%), breeding-site observations in 20 studies (n = 20, 32.8%), and physical specimen submissions (eggs, larvae, or adult mosquitoes) in 15 studies (n = 15, 24.6%) [12,21,26–31,35,36,43,45,52,58].

**3.6.2. Platforms and technologies.** A wide set of platforms and technologies supported these data streams (Fig 4A). Mobile applications and web portals were used in 37 studies (n = 37, 60.7%). Among these, Mosquito Alert (https://www.mosquitoalert.com/en/) was the most frequently used platform (n = 14, 23.0%) [2,7,9,11,19–24,34,35,37,42], followed by GLOBE Observer (https://observer.globe.gov/) (n = 8, 13.1%) [10,11,39,42,59,70,71], iNaturalist (https://www.inaturalist.org/) (n = 7, 11.5%) [11,38,42,63–65,72], and Mückenatlas web portal (https://mueckenatlas.com/) (n = 6, 9.8%) [26–31].

Other mobile applications were reported less frequently and included ZanzaMapp (https://www.zanzamapp.it/) (n = 2, 3.3%) [32,33], MozzWear for acoustic detection (n = 2, 3.3%) [54,55], Mo-Buzz for community reporting (n = 1, 1.6%) [69], EntoLab for expert-assisted validation (n = 1, 1.6%) [2], Meditrack (n = 1, 1.6%) [66], and Trip Doctor (n = 1, 1.6%) [25]. One study developed an educational web portal ("Preparedness") (https://www.azdhs.gov/preparedness/) designed for teachers and youth leaders to introduce mosquito biology (n = 1, 1.6%) [43].

Geolocation technologies, including GPS-based location capture, were used to enrich spatial precision in 14 studies (n = 14, 23.0%) [2,7,10,11,19,22–24,39,41,42,51,66,69]. Sensor-based technologies were used in a limited subset of studies (n = 6, 9.8%), including larval monitoring devices or IoT-enabled larval sensors (n = 3, 4.9%) [42,66,68] and acoustic sensors for detecting mosquito flight tones (n = 3, 4.9%) [41,54,55]. Smart-trap hardware, including BG-GAT and BG-Mosquitaire systems, was used in one study (n = 1, 1.6%) [12], macro lenses to enhance image resolution were also

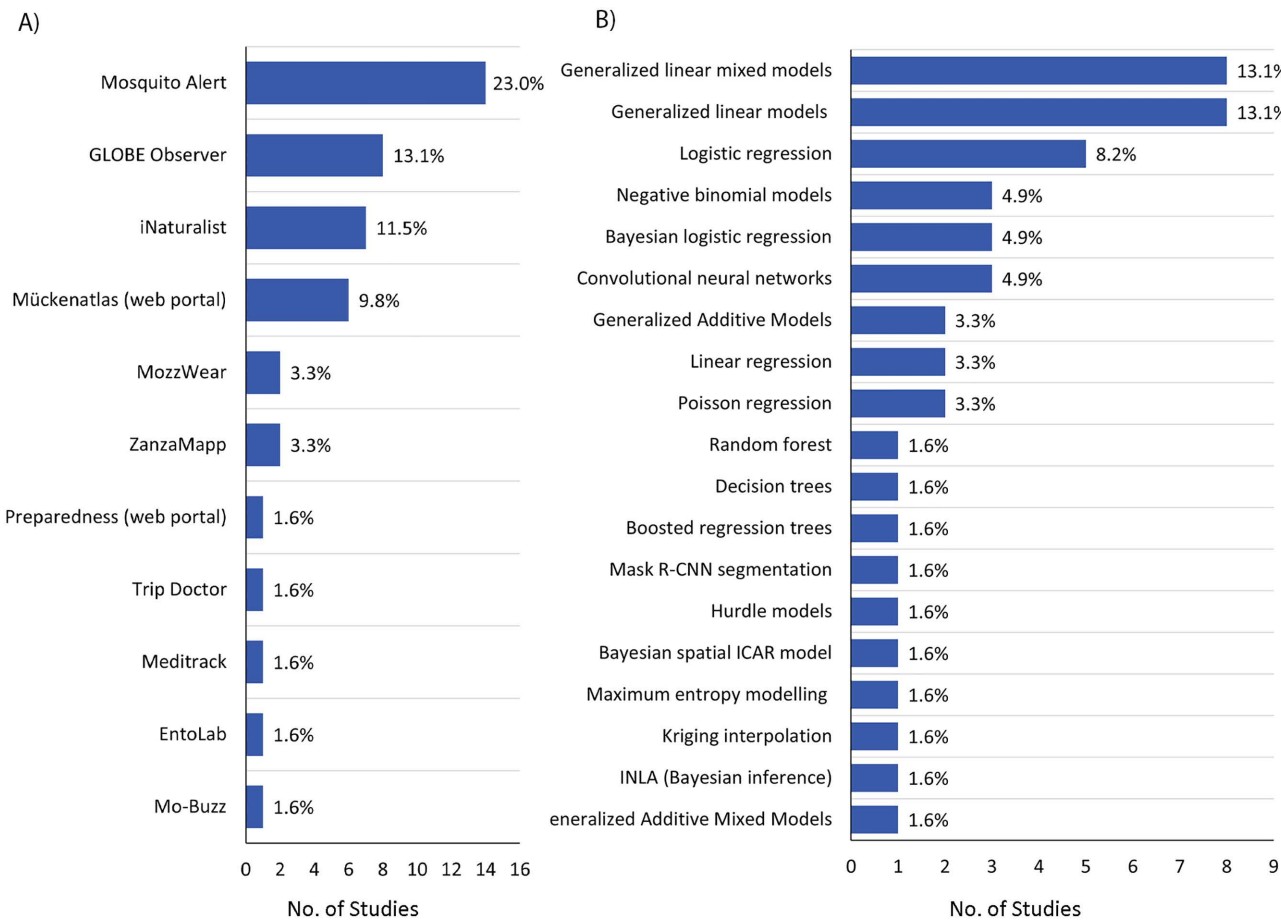

**Fig 4. Citizen science (A) platforms and applications, and (B) analytical approaches used in MBDs surveillance and control across included studies.**

reported in one study (n = 1, 1.6%) [11]. In general, smartphone- and web-based platforms enabled scalable, geolocated reporting across wide areas. In contrast, sensor-based, trap-based, or educational tools were more often tied to specialized surveillance tasks, local monitoring, or training-oriented participation models.

### 3.7. Validation mechanisms

Validation of citizen-generated observations was addressed in 48 studies (n = 48, 78.7%). In the remaining 13 studies (n = 13, 21.3%), validation was not applicable because the primary outcomes were derived from independent entomological monitoring or intervention assessments (n = 9, 14.8%), conceptual or educational program design (n = 3, 4.9%), or non-vector reporting systems (n = 1, 1.6%), rather than from citizen-submitted records requiring taxonomic verification.

Expert-led validation was the dominant approach (n = 33, 54.1%). These workflows included app- or photo-based expert review (n = 16, 26.2%), laboratory processing of mailed specimens or ovitrap materials (n = 15, 24.6%), and expert validation followed by field confirmation (n = 2, 3.3%). Morphological examination, rearing, or molecular confirmation supported expert-led workflows in 11 studies (n = 11, 18.0%).

Hybrid or mixed validation approaches were reported in six studies (n = 6, 9.8%), typically combining automated classification, participant-assisted identification, or cross-platform workflows with subsequent expert review. Fully automated classification was uncommon (n = 3, 4.9%) and largely limited to acoustic detection systems. Other validation approaches appeared only once each (n = 1, 1.6%), including community-based classification, participant self-identification supported by voucher photographs, and independent entomological verification of community-ranked mosquito-density outputs. In three studies (n = 3, 4.9%), validation procedures were not clearly described.

Validation approaches varied systematically by platform type. App- and image-based reporting systems most often relied on expert review or hybrid workflows, whereas specimen- and trap-based systems commonly used laboratory or field confirmation, reflecting differences in data type and validation demands created by platform design.

Quantitative validation performance was reported inconsistently. Five studies (n = 5, 8.2%) reported expert–participant agreement or image-identification accuracy ranging from approximately 86% to 94%. One study (n = 1, 1.6%) reported species-specific expert-confirmed accuracies of 87.6% for *Aedes albopictus* and 81.3% for *Aedes aegypti*, another (n = 1, 1.6%) reported early-warning specificity of approximately 97%, and one (n = 1, 1.6%) reported a ROC AUC of about 0.96 with approximately 98% accuracy for a large subset of images. Validation performance was more variable in one larval self-identification study (n = 1, 1.6%), with accuracy ranging from 34% to 100% across countries, and an acoustic-classification study (n = 1, 1.6%), where accuracy improved from approximately 35% to 65% after incorporating location metadata.

### 3.8. Analytical approaches, effort-bias handling, and epidemiologic outputs

Analytical approaches varied according to the type of citizen-generated data, the surveillance objective, and the extent to which studies addressed validation and uneven reporting efforts. These approaches included traditional statistical models, ecological models, machine learning (ML), and spatially explicit analyses. Generalized linear models (GLMs) were applied in eight studies (n = 8, 13.1%) [12,32–34,40,52–54], similar to generalized linear mixed models (GLMMs) (n = 8, 13.1%) [9,12,34,37,40,48,49,58]. These approaches were used to analyze citizen-submitted mosquito observations, model associations with environmental drivers, and support species-identification workflows.

Logistic regression models were applied in five studies (n = 5, 8.2%) [20,27,32,42,56], whereas Bayesian logistic regression featured in three studies (n = 3, 4.9%) [7,20,23]. Negative binomial models were employed in three studies (n = 3, 4.9%) [12,32,56], Poisson models in two studies (n = 2, 3.3%) [23,34], and linear regression in two studies (n = 2, 3.3%) [33,42]. Hurdle models appeared in only one study (n = 1, 1.6%) [30].

Generalized Additive Models (GAMs) were applied in two studies (n = 2, 3.3%) [9,33], with a generalized additive mixed model (GAMM) used in one study (n = 1, 1.6%) [32]. More specialized analytic methods were reported infrequently, including Bayesian inference using the Integrated Nested Laplace Approximation framework (n = 1, 1.6%) [32], spatial

interpolation techniques using kriging (n = 1, 1.6%) [32], species distribution modeling using maximum entropy modeling (n = 1, 1.6%) [59] and Bayesian spatial models based on intrinsic conditional autoregressive structure (n = 1, 1.6%) [23].

ML approaches were identified in six studies (n = 6, 9.8%) [11,18,24,35,42,60], primarily to support automated classification and species identification from citizen-contributed data. These included convolutional neural networks (CNNs) (n = 3, 4.9%) [11,18,24], Mask R-CNN segmentation (n = 1, 1.6%) [11], boosted regression trees (n = 1, 1.6%) [35], decision trees (n = 1, 1.6%) [42], and random forest (n = 1, 1.6%) [42]. Fig 4B summarizes the most common analytical approaches used across included studies.

Handling of participation or reporting effort varied substantially across the included studies. In 23 studies (n = 23, 37.7%), effort correction was not directly applicable because the analyses focused on intervention trials or structured entomological monitoring (n = 9, 14.8%), co-design, implementation, or conceptual work (n = 5, 8.2%), participation- or questionnaire-based non-ecological analyses (n = 4, 6.6%), educational outcomes (n = 2, 3.3%), image-classification studies (n = 2, 3.3%), or non-vector reporting systems (n = 1, 1.6%).

Explicit analytical adjustment or effort-aware modeling was identified in 11 studies (n = 11, 18.0%). These included offsets for estimated sampling effort (n = 2, 3.3%), weighted pseudo-absences or effort-weighted absences (n = 2, 3.3%), effort-standardized abundance metrics (n = 2, 3.3%), reporting-propensity models (n = 2, 3.3%), and direct modeling of reporting behavior (n = 1, 1.6%) (non-mutually exclusive).

In 10 studies (n = 10, 16.4%), participation bias was examined, benchmarked, or partly accommodated without direct correction. These studies benchmarked citizen science records against external surveillance or reference datasets (n = 4, 6.6%), filtered low-quality or anomalous records (n = 1, 1.6%), or qualitative discussion of population-related reporting bias (n = 9, 14.8%) (non-mutually exclusive). A further 10 studies (16.4%) focused on detection-oriented outputs (e.g., early detection, confirmed occurrence, range expansion) and did not apply formal effort correction. In seven studies (n = 7, 11.5%), effort-related bias was partly mitigated through structured or standardized citizen-assisted sampling designs without separate analytical correction.

The impact of uneven reporting varied by outcome. Outputs based on species presence, first detection, or confirmed occurrence were generally less sensitive to uneven reporting intensity. In contrast, abundance estimates, hotspot mapping, spatial clustering, temporal trend analyses, and some risk-modeling outputs were more vulnerable to participation bias. Accordingly, the implications of effort bias varied across studies depending on the type of surveillance output generated.

Platforms based on low-burden public reporting, often supported by expert validation, were commonly used for species occurrence mapping, early detection, and tracking of spread, particularly for invasive mosquitoes. Studies that produced more quantitative outputs, such as abundance estimates, hotspot mapping, temporal trend analysis, or risk modeling, were more likely to incorporate stronger validation workflows, structured sampling, external benchmarking, or explicit effort-aware modeling. Community-based initiatives with deeper local engagement were more frequently associated with education, intervention, and locally grounded monitoring, even when they generated fewer analytically intensive outputs. These roles often overlapped, and several studies combined surveillance, engagement, and control objectives.

Predictive and epidemiological outputs varied across studies but clustered into several recurring categories. Trend analyses were the most common outputs (n = 32, 52.5%). These analyses characterized seasonal MBDs activities (n = 15, 24.6%) [20,22,26,28,34,35,38,41,54,55,58,59,66,67,72], population trends (n = 7, 11.5%) [10,25,28,57,58,64,70], spatio-temporal submission patterns (n = 4, 6.5%) [2,9,30,58], ecological or behavioral changes over time (n = 3, 4.9%) [9,30,59,70], and intervention effects (n = 1, 1.6%) [36].

Risk-factor identification appeared in 27 studies (n = 27, 44.3%). These studies examined environmental (n = 12, 19.7%) [2,10,12,23,27,41,51–53,56,58,69,70], climatic (n = 5, 8.2%) [22,33,46,60,72], demographic or behavioral (n = 3, 4.9%) [9,27,52], and land cover predictors (n = 1, 1.6%) [10].

Spatial predictions, including distribution mapping and ecological suitability modeling, were documented in 26 studies (n = 26, 42.6%). These outputs included species distribution maps (n = 13, 21.3%) [2,7,10,22,26,28,31,35,37,38,52,59,72], expansion or spread predictions (n = 5, 8.2%) [2,23,41,42,71], ecological suitability models (n = 4, 6.6%) [7,9,10,41], multi-year spatial trend maps (n = 2, 3.3%) [32,66], and spatial interpolation techniques (n = 1, 1.6%) [52].

Hotspot mapping was reported in 12 studies (n = 12, 19.7%). These outputs visualized density-based clusters (n = 5, 8.2%) [33,46,52,69,71], ecological hotspots (n = 3, 4.9%) [22,42,69], breeding-site aggregations (n = 2, 3.3%) [37,71], and early-warning hotspot visualizations (n = 2, 3.3%) [42,60].

Risk modeling outputs were documented in 10 studies (n = 10, 16.4%). These studies included exposure risk estimates (n = 5, 8.2%) [2,23,35,42,69], outbreak potential assessments (n = 3, 4.9%) [41,42,69], and species-encounter probability surfaces (n = 2, 3.3%) [7,23].

### 3.9. Reported biases and methodological limitations

Across the included studies, several explicitly reported biases or methodological limitations affecting their own citizen science data or implementation. Data quality issues were the most common (n = 35, 57.4%), followed by validation limitations (n = 32, 52.5%), recruitment/engagement challenges (n = 32, 52.5%), and participation and sampling biases reported by the studies themselves (n = 27, 44.3%), which included uneven observation or reporting effort. Representativeness limitations, indicating that the contributed data may not adequately reflect the full target population or study context, were noted in 25 studies (n = 25, 41.0%). Technology-related barriers (n = 16, 26.2%) and spatial bias (n = 15, 24.6%) were reported less often.

### 3.10. Data governance and participant protections

Ethical practices varied across the included studies. Informed consent was explicitly reported in 34 studies (n = 34, 55.7%). Consent mechanisms included written agreements in community or school-based settings (n = 8, 13.1%) [44,48,49,53–57] and digital or implicit consent obtained through mobile-app user agreements or platform participation (n = 8, 13.1%) [2,10,22,34,35,37,38,59]. Privacy protections were described in 32 studies (n = 32, 52.5%).

Ethics committee or institutional review board approval was reported in 11 studies (n = 11, 18.0%) [40,44,47–50,52–55,57], primarily in studies involving human subjects, minors, biological specimen collection, or community-based intervention components. Data ownership and licensing considerations were reported in 14 studies (n = 14, 23.0%) [7,9–11,22,31,32,35,38,41,42,60,64,70]. These studies specified user ownership of submitted content [64] or outlined open-data policies such as Creative Commons licensing (e.g., CC0) under platforms such as Mosquito Alert [22]. Notably, several recurring digital platforms operated under established governance frameworks that commonly addressed consent, privacy, anonymization, visibility, and data sharing. As these provisions were often embedded within the platform itself, they were not always restated explicitly in each study using the same tools. For example, Mosquito Alert incorporated user agreement and privacy policy acceptance, anonymous participation, random user identifiers, geolocation protection, and open data or licensing terms. GLOBE Observer used account-based participation with public screen names (rather than public personal identifiers) and screened submitted photographs for faces or identifying text. Mückenatlas used explicit consent procedures, confidential handling of personal data, anonymized sharing of mosquito and location information, and opt-in public visibility. Similarly, iNaturalist allowed users to retain ownership of submitted content while selecting licenses and controlling geoprivacy settings for observations.

## 4. Discussion

This review provided a comprehensive synthesis of how citizen science has been applied to MBDs surveillance and control over the past two decades. The evidence base has expanded rapidly since 2017, reflecting growing interest in participatory, digitally enabled surveillance approaches and the maturation of mobile technologies that facilitate large-scale

 

public engagement. Across studies, citizen science has been used most consistently to support mosquito monitoring, particularly for invasive and nuisance species. It has generated a range of surveillance-relevant outputs, including spatio-temporal trend analyses, distribution mapping, hotspot identification, and risk-related products. However, the literature points to persistent gaps that limit translation into public health decision-making, including limited evaluation of surveillance system performance, fewer intervention-oriented studies, incomplete reporting of participant characteristics, and limited data governance and ethical safeguards [11].

Our findings confirm and extend prior reviews showing that citizen science can complement routine entomological surveillance by expanding geographic coverage and supporting early detection, particularly for invasive *Aedes* species. Consistent with earlier work, we also observed substantial methodological heterogeneity, uneven representativeness, and ongoing needs for robust validation workflows to enhance public health utility [13]. Our review advances the literature by providing a more integrated synthesis across the surveillance continuum, explicitly linking recruitment strategies and participation objectives to citizen-generated data streams, validation workflows, analytical approaches, and resulting epidemiologic outputs. By identifying where evidence concentrates, we can clarify how citizen-generated data are used in surveillance practice and where methodological improvements are needed to enhance public health impact. The review further suggests that the scientific value of citizen-generated mosquito data depends not only on the underlying validation workflow, but also on how the participation model, effort-bias handling, and analytical objective are aligned. Low-burden public-reporting systems with strong expert validation were most often used for early detection, occurrence mapping, and spread tracking. In contrast, studies targeting quantitative outputs, such as hotspot mapping, temporal trend analysis, or risk modeling, more often require stronger validation, benchmarking, and effort-aware analytical approaches. Community-based initiatives with deeper local engagement were more often oriented toward education, public engagement, and local vector-control activities. These roles often overlapped rather than being mutually exclusive. In this sense, citizen science can be understood as ground-truth, human-sensed, and socially mediated geospatial data that extend surveillance reach and timeliness while also reflecting the social geography of participation and associated reporting biases. This framing is particularly relevant during crises or operational disruptions, when routine field surveillance may be delayed or unable to scale and citizen-generated observations can provide rapid situational awareness [73].

The geographic distribution of studies was uneven, with most evidence originating from Europe and the Americas and comparatively limited representation from Asia and parts of Africa. This pattern likely reflects differences in research infrastructure, sustained funding, access to mature digital platforms capable of supporting large-scale participation, and region-specific surveillance priorities. In Europe, many studies have focused on invasive *Aedes* detection and monitoring, where citizen reporting is particularly well aligned with early detection and spread tracking [22,72]. This pattern is likely further influenced by variation in regulatory and data-sharing environments, unequal academic incentives to publish citizen science research, and language or indexing biases that can obscure locally implemented surveillance efforts.

Our findings point to a meaningful gap in how participant characteristics and contextual factors are reported and examined in citizen science–based MBD surveillance studies. Gender distributions were documented in only a small minority of studies, and implementations focused exclusively on rural settings were relatively uncommon. These omissions are not merely descriptive limitations; they have direct implications for data quality, representativeness, and equity [74,75]. Participation in citizen science is shaped by structural factors such as access to digital technologies, educational background, and institutional trust, all of which can influence who contributes data and where observations are generated [76]. Consequently, citizen-generated datasets often overrepresent urban, digitally connected populations while under-capturing rural and underserved communities. These biases may produce systematic blind spots in populations that frequently experience higher vulnerability to MBDs and limited coverage by routine surveillance systems. Consistent with this pattern, only a small subset of studies implemented clearly targeted recruitment strategies for rural, remote, or underserved settings. Most addressed this issue only indirectly through broader outreach or did not clearly report a recruitment-side strategy to mitigate structural participation imbalance.

Across the included studies, several recurring limitations constrained both the interpretability and operational utility of citizen science–based surveillance. Because participation is voluntary and geographically uneven, data are often clustered in accessible or highly engaged areas, creating spatial imbalances that complicate interpretation [77]. Data quality challenges, including misidentification, variable image quality, and incomplete submissions, further complicated analysis and interpretation [78]. A persistent methodological issue was the difficulty of disentangling reporting effort from true mosquito abundance, particularly when submission counts were treated as implicit proxies of population levels. Only a minority of studies used explicit effort-aware modeling. Most either addressed participation bias qualitatively or focused on outputs less sensitive to uneven reporting, such as early detection or confirmed presence. This distinction is important because abundance estimates, hotspot mapping, spatial clustering, and temporal trend analyses are more vulnerable to participation bias than detection-oriented surveillance outputs. From an implementation perspective, sustaining participant engagement over time and maintaining adequate validation capacity emerged as major constraints, especially for systems that rely on expert review for species confirmation or for curating training data for automated classification. While these constraints do not negate the value of citizen science for MBDs surveillance, they underscore the need for clearer reporting of effort and validation, and for routine benchmarking against entomological data where feasible.

Another related methodological priority is uncertainty propagation in citizen science–based mosquito surveillance. Uncertainty may arise from observation bias, when reports reflect where people are present rather than where mosquitoes are distributed [79]. It can also stem from effort bias, as submission counts are not equivalent to mosquito abundance [80]. Additional sources of uncertainty come from classification error, particularly in image-, acoustic-, and machine-learning–based identification [78,81]. Furthermore, spatial aggregation choices can blur the signal and make it difficult to separate true hotspots from reporting artefacts [82]. These challenges point to the need for effort-aware probabilistic approaches, including Bayesian frameworks, that represent and propagate uncertainty across the surveillance pipeline instead of treating citizen observations as error-free inputs [83].

Our synthesis suggests that many included studies primarily reflect an extractive "citizens as sensors" model, in which participation is dominated by data provision. At the same time, emerging evidence of Human-in-the-Loop participation, including dataset annotation and limited citizen involvement in data analysis, indicates opportunities to move toward more participatory models. This aligns with typologies of participation that distinguish crowdsourcing from deeper forms of participatory science, and underscores the value of explicitly designing citizen science systems that incorporate validation loops, feedback, and shared interpretation [84]. Future mosquito surveillance initiatives could more consistently position participants as active analytical partners by integrating citizen contributions into model refinement, data validation loops, and iterative improvement of classification workflows, rather than limiting citizen roles to data submission alone.

Ethical and privacy considerations were described unevenly across the literature. While some studies reported informed consent, privacy protections, or formal ethics review, many provided limited or no details on these practices. This variability likely reflects differences in how projects conceptualize citizen science, as environmental monitoring rather than human-subjects research, as well as reliance on app-based terms of use in place of explicit ethical protocols. Beyond reporting practices, ethical transparency plays a central role in citizen science systems by shaping participant trust, engagement, and sustained contribution. Because these initiatives rely on voluntary public participation, clear communication about what data are collected, how they are used, how privacy is protected, and whether submissions are anonymized or shared openly is not only an ethical requirement but also a practical mechanism for encouraging participation. Many mosquito-focused citizen science platforms deliberately minimized personal data collection, used anonymized or masked identifiers, or allowed users to control visibility and sharing settings. When clearly communicated, these protections may therefore strengthen confidence in participation and support sustained data contribution. In this context, transparent and standardized reporting of governance practices would provide a better understanding of how citizen science systems foster trust while managing privacy, data use, and accountability. Nonetheless, citizen science–based MBDs surveillance often involves precise geolocation and time-stamped data, which can pose privacy risks, particularly when

information is shared openly or visualized at fine spatial scales [7]. Established ethical frameworks for participatory and digital surveillance emphasize explicit consent, data minimization, clear governance, and fairness [85]. Without systematic reporting and consistent implementation of these safeguards, the credibility, trustworthiness, and long-term sustainability of citizen science initiatives may be undermined [86]. To move beyond identifying ethical gaps and toward actionable guidance, future studies should report governance practices more explicitly. This includes clarifying data sovereignty by specifying who owns and controls citizen-generated data, and outlining benefit-sharing mechanisms, such as whether participating communities receive timely results and actionable outputs. Studies should also strengthen informed consent for fine-scale geolocation sharing, with clear options for spatial generalization and withdrawal. In addition, transparency around AI-assisted validation and classification is essential, including monitoring for potential algorithmic bias. Reporting these elements as standard practice would strengthen governance and support ethical, sustainable deployment in public health settings.

Several limitations meaningfully constrain the current evidence base and its operational relevance. Restricting the review to English-language, peer-reviewed studies indexed in major databases likely underrepresents work published in other languages, reported through program evaluations, or disseminated as gray literature, particularly from regions where citizen science operates outside traditional academic channels. Inconsistent terminology and definitions of citizen science across disciplines further complicate synthesis, while substantial heterogeneity in study designs, reporting practices, and outcome measures limits comparability and limits quantitative analysis. Addressing these challenges is essential to advancing the field. Future research should prioritize longitudinal and evaluation-oriented designs that quantify timeliness, accuracy, and added value relative to routine surveillance, incorporate intervention studies that assess decision-making and risk reduction, strengthen integration with contextual and social data and remote sensing linkages, and adopt consistent reporting standards for participation, validation, uncertainty, and ethics to support reproducibility and public health translation.

## 5. Conclusions

This review showed that citizen science has become an increasingly important complement to conventional mosquito surveillance, particularly for monitoring invasive species and generating timely, spatially resolved outputs such as trend analysis, distribution mapping, and hotspot identification. However, uneven geographic coverage, participation biases, variable data quality, limited validation, and inconsistent ethical reporting continue to limit operational uptake. Addressing these challenges through rigorous evaluation, standardized reporting, and clear governance frameworks will be essential for translating citizen science into reliable and sustainable mosquito surveillance and control.

## Supporting information

**S1 File. Preferred Reporting Items for Systematic reviews and Meta-Analyses extension for Scoping Reviews (PRISMA-ScR) Checklist.**
(DOCX)

**S2 File. Detailed extracted information of the included articles.**
(DOCX)

## Author contributions

**Conceptualization:** Nima Kianfar, Kimia Savoji, Xiao Huang, Di Yang, Abe Mollalo.

**Formal analysis:** Nima Kianfar, Kimia Savoji.

**Funding acquisition:** Xiao Huang, Di Yang, Abe Mollalo.

**Software:** Nima Kianfar, Kimia Savoji, Abe Mollalo.

**Supervision:** Abe Mollalo.

**Writing – original draft:** Nima Kianfar, Kimia Savoji, Abe Mollalo.

**Writing – review & editing:** Nima Kianfar, Xiao Huang, Di Yang, Abe Mollalo.

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
