## [Decision Letter · Decision Letter 0]

19 Mar 2026

PONE-D-26-07716The Role of Citizen Science in Mosquito-Borne Disease Surveillance and Control: A Scoping ReviewPLOS One

Dear Dr. Mollalo,

Thank you for submitting your manuscript to PLOS ONE. After careful consideration, we feel that it has merit but does not fully meet PLOS ONE’s publication criteria as it currently stands. Therefore, we invite you to submit a revised version of the manuscript that addresses the points raised during the review process.

We look forward to receiving your revised manuscript.

Kind regards,

Luca Nelli, PhD

Academic Editor

PLOS One

Additional Editor Comments:

Dear Abe Mollalo,

Thank you for submitting your manuscript to PLOS One.

After editorial assessment and external review, I am writing to let you know that the decision is major revision. The reviewers and I feel that the topic is relevant and the manuscript has clear potential, but several substantive issues need to be addressed before it can be considered further.

Please revise the manuscript carefully in light of the reviewers’ comments and provide a clear, point-by-point response explaining how each issue has been addressed.

I look forward to receiving your revised submission.

Best wishes,

Luca Nelli

Reviewers' comments:

Reviewer's Responses to Questions

**Comments to the Author**

1. Is the manuscript technically sound, and do the data support the conclusions?

Reviewer #1: Yes

Reviewer #2: Partly

2. Has the statistical analysis been performed appropriately and rigorously? 

Reviewer #1: Yes

Reviewer #2: N/A

3. Have the authors made all data underlying the findings in their manuscript fully available?

Reviewer #1: Yes

Reviewer #2: Yes

4. Is the manuscript presented in an intelligible fashion and written in standard English?

Reviewer #1: Yes

Reviewer #2: Yes

5. Review Comments to the Author

Reviewer #1: The paper provides a scoping review of studies incorporating citizen science elements for mosquito surveillance. The work offers a useful framework through which to understand global variation and biases in citizen science data within the field of mosquito-borne diseases (MBDs). The authors provide clear rationale for the work and the inclusion criteria are justified. The analysis is comprehensive and includes many parameters to identify differences and commonalities of studies. Statistical analysis is limited to percentages but is appropriate for the objectives. The authors find biases in spatial distribution of studies, with the majority of studies occurring in the western hemisphere and the Global North. The work adds to existing literature on the topic of citizen science for vector surveillance through the inclusion here of ethical considerations and participation models, as well as useful discussion of spatial and methodological biases. The paper is written well with clear objectives, methodology, findings and conclusions. All supporting data are available within the paper and supplementary information.

Minor comments:

Line 172-181: Listing the numbers of countries that contributed studies from each continent may cause confusion as this is not reflected in the map in Figure 2, where far more countries are included due to multi-country studies - this is made clear in the supporting information but is not explicit in the manuscript. It would be helpful to mention the total number of countries from which citizen science studies were included, either in the same paragraph or within the figure legend for Figure 2.

Line 194-195: It might be useful to clarify how species were determined to be invasive – was this based on studies that specifically mentioned invasive species monitoring? If not, was the definition of ‘invasive’ accounted for across studies that may have included the native range of ‘invasive’ species e.g. Aedes aegypti?

Line 211: Maybe the number of studies from the Americas and Oceania that included surveillance for dengue/ Zika/ Chikungunya could also be provided, for consistency

Line 240: It would be interesting to know what other traps were used and the breakdown of trap types by number of studies

Line 271-272: Maybe the last three approaches can be reordered to reflect decreasing frequency (public websites, social media campaigns, institutional newsletters)

General comment: The discussion lists several biases of citizen science data. While not necessary, it could be interesting to give an overview (in the results) of biases that the studies themselves have identified in their work.

Reviewer #2: This manuscript presents a scoping review examining the role of citizen science in mosquito-borne disease surveillance and control. The authors analyze 61 studies published between 2000 and 2025 that use citizen participation to collect mosquito-related data or support vector surveillance initiatives. The review summarizes study characteristics, technologies used, participation strategies, analytical outputs, and governance practices. The topic is timely and highly relevant. Citizen science initiatives have become an increasingly important complement to traditional mosquito surveillance programs, particularly with the widespread availability of mobile technologies and digital reporting platforms. The manuscript provides a useful overview of existing initiatives and highlights the rapid growth of citizen-science projects in this field. However, several aspects that determine the scientific robustness and operational value of citizen-science surveillance systems are insufficiently addressed in the analysis and would benefit from further development.

A first aspect concerns the discussion of ethical and governance issues. In the current manuscript, governance considerations are mainly addressed in terms of whether ethical procedures are reported by the studies included in the review. While this is important, the discussion could be expanded to consider the role of ethical transparency as a mechanism to foster participation and trust among citizens. Citizen-science initiatives depend fundamentally on voluntary engagement from the public, and participation levels are strongly influenced by the degree of trust participants place in the project. Clear communication about how data are collected, how they will be used, and how personal information is protected is, therefore, not only an ethical requirement but also a practical mechanism to encourage participation. In many mosquito citizen-science platforms, personal data collection is deliberately minimized or avoided, and the explicit communication of this principle can serve as an important incentive for participation. Addressing how different projects communicate these governance and privacy practices to participants would therefore provide a more nuanced perspective on the role of ethics and governance in citizen-science initiatives. Linked to that, as the manuscript highlights, many studies are related to the same citizen science project or use the same platform, and authors should make sure that in all these cases, some basic information related to ethical or data collection issues should not be mentioned, considering that it has been previously described or explained in previous papers. Saying that to make note of the authors to check for consistency for platforms or projects beyond what is explained or described in a single paper. For example, checking Supplementary file 2 “Extraction table” it is possible to see that kind of inconsistencies between study 1 and 2, both use Mosquito Alert tools, in the 1st ethics about the obtetion of pictures and reports is stated (e.g. user agreement, anonymous participation, etc.), while in the 2nd, even if also use Mosquito Alert for image classification, the obtention of the images appears as “Not stated”. If they used the Mosquito Alert app to collect images, it is supposed that reports have been collected with user agreement and anonymous participation again. I will recommend the authors in that table to create a clear field identifying the project or tool used to unify the studies and check that information is consistent across all studies for the same platform and tool.

A second important issue relates to participation biases in citizen-science datasets. A well-known limitation of citizen-science initiatives is that participation intensity is strongly dependent on population density. Areas with larger populations tend to generate more observations simply because more potential contributors are present, whereas rural or sparsely populated areas often remain underrepresented. This urban participation bias has important implications for mosquito surveillance systems, particularly when citizen-generated data are used for spatial analyses such as hotspot detection, abundance estimation, or risk mapping. The manuscript notes that a large proportion of studies were conducted in urban environments, and in the discussion highlights the bias risk, but the analysis would benefit from exploring more explicitly how different projects address this structural bias. Some initiatives attempt to mitigate these imbalances through targeted recruitment strategies, for example, by organizing communication campaigns or partnerships aimed at engaging communities in rural or less populated regions where participation is otherwise more difficult to achieve. Examining whether and how such strategies are implemented across the reviewed studies would provide valuable insight into how citizen-science projects attempt to balance participation across different geographical contexts.

Closely related to this issue is the analytical treatment of participation biases in the studies reviewed. Citizen-science datasets inevitably reflect both ecological patterns and patterns of reporting activity, and this distinction is particularly important when data are used to generate quantitative outputs. Analyses based on citizen observations should ideally account for variations in participation effort, as differences in reporting intensity can strongly influence estimates of mosquito abundance, the identification of spatial hotspots, or the interpretation of temporal trends. Without appropriate corrections, such outputs may reflect patterns of participation rather than true ecological dynamics. The manuscript would therefore benefit from a more systematic examination of how the reviewed studies address sampling effort in their analyses. Some analytical approaches explicitly incorporate participation intensity or observation probability, whereas others use modelling frameworks designed to account for detection biases. At the same time, it is important to recognize that not all types of outputs are equally sensitive to participation bias. For example, detecting the presence of a mosquito species in a region is relatively independent of sampling effort, as the species is either detected or not detected. In contrast, estimates of abundance, spatial clustering, or temporal trends are much more sensitive to variations in participation effort. Distinguishing between these different types of outputs and discussing how they are affected by participation biases would significantly strengthen the analytical depth of the review.

Another important dimension that appears to be largely absent from the current analysis concerns the validation of citizen-generated observations. The scientific reliability of citizen-science mosquito surveillance depends critically on how submitted observations are verified and classified. Different projects employ a range of validation mechanisms, including expert entomologist review, distributed networks of specialists, community validation processes, or automated classification tools based on artificial intelligence. These procedures can vary considerably in terms of accuracy, speed, and scalability, and they directly influence the reliability of the datasets generated by citizen-science platforms. The review would therefore benefit from examining how the different initiatives included in the study handle the classification and validation of mosquito specimens or images, who performs these validations, and what is known about the accuracy of the identification processes. Understanding these mechanisms is essential for assessing the scientific value of the outputs generated by the platforms.

More broadly, and more importantly, from my point of view, the manuscript would benefit from adopting a more integrated analytical framework linking the different components of citizen-science systems. At present, the review discusses aspects such as data collection, participation strategies, analytical outputs, and governance largely as separate dimensions. However, the scientific value of a citizen-science platform ultimately depends on how these elements interact. A more synthetic perspective that connects the level of citizen participation, the validation procedures used, the analytical approaches applied, and the methods used to correct participation biases would provide a clearer picture of the strengths and limitations of different initiatives. Such an approach could also help identify distinct functional roles of citizen-science projects. Some platforms may primarily contribute to quantitative public-health surveillance by generating data that can be integrated into risk models, while others may focus more strongly on education and public engagement or on supporting early detection of invasive species. These different functions are not mutually exclusive, and no single model should necessarily be considered superior to others. However, distinguishing between them could help clarify the specific contributions that citizen-science initiatives can make to mosquito surveillance and public health. Developing this type of synthesis would significantly strengthen the manuscript by offering a more comprehensive overview of the diversity of citizen-science platforms and the different roles they play.

In summary, the manuscript addresses an important and rapidly evolving area of research and provides a useful overview of existing citizen-science initiatives in mosquito-borne disease surveillance. However, the review would benefit from a deeper analysis of participation biases, validation procedures, and analytical approaches used to account for variations in participation effort. Expanding the discussion of governance and ethical transparency in relation to citizen participation, and developing a more integrated framework linking participation, validation, analysis, and outputs, would greatly enhance the contribution of the study. With these revisions, the manuscript could provide a more robust and informative synthesis of the current state of citizen-science approaches in mosquito surveillance.

6. PLOS authors have the option to publish the peer review history of their article (what does this mean?). If published, this will include your full peer review and any attached files.

Reviewer #1: No

Reviewer #2: No

---

## [Author Response · Author response to Decision Letter 1]

12 Apr 2026

Dear Editor and Reviewers,

Thank you for your time and careful review of our manuscript. We have thoroughly revised the manuscript in response to your valuable comments. We believe the revised version is substantially improved in clarity, organization, and presentation. Below, we provide a point-by-point response to each comment and indicate where revisions were made in the manuscript.

Sincerely,

Response to Editor

COMMENT

After editorial assessment and external review, I am writing to let you know that the decision is major revision. The reviewers and I feel that the topic is relevant and the manuscript has clear potential, but several substantive issues need to be addressed before it can be considered further.

Please revise the manuscript carefully in light of the reviewers’ comments and provide a clear, point-by-point response explaining how each issue has been addressed.

RESPONSE

Thank you for your guidance and for the opportunity to revise our manuscript. We have carefully addressed reviewers’ comments and provided a detailed point-by-point response explaining how each issue was handled. All changes are highlighted in the revised manuscript.

Responses to Reviewer #1

Comments to the Author

The paper provides a scoping review of studies incorporating citizen science elements for mosquito surveillance. The work offers a useful framework through which to understand global variation and biases in citizen science data within the field of mosquito-borne diseases (MBDs). The authors provide clear rationale for the work and the inclusion criteria are justified. The analysis is comprehensive and includes many parameters to identify differences and commonalities of studies. Statistical analysis is limited to percentages but is appropriate for the objectives. The authors find biases in spatial distribution of studies, with the majority of studies occurring in the western hemisphere and the Global North. The work adds to existing literature on the topic of citizen science for vector surveillance through the inclusion here of ethical considerations and participation models, as well as useful discussion of spatial and methodological biases. The paper is written well with clear objectives, methodology, findings and conclusions. All supporting data are available within the paper and supplementary information.

RESPONSE

We sincerely thank Reviewer #1 for their time and valuable feedback. Their insightful comments have significantly contributed to improving the quality of our manuscript. In this revised version, we have provided specific responses to each comment and highlighted all corresponding changes in the text.

**

COMMENT

Minor comments:

Line 172-181: Listing the numbers of countries that contributed studies from each continent may cause confusion as this is not reflected in the map in Figure 2, where far more countries are included due to multi-country studies - this is made clear in the supporting information but is not explicit in the manuscript. It would be helpful to mention the total number of countries from which citizen science studies were included, either in the same paragraph or within the figure legend for Figure 2.

RESPONSE

Thank you for this helpful suggestion. We agree that this point required clarification and have revised both the Results section 3.2 and the Figure 2 caption accordingly:

Added text to Results section 3.2:

“Overall, the included studies covered 24 unique countries.”

Added text to Figure 2 caption:

“Panel A shows all countries represented across the included studies (n=24 unique countries), including those captured through multi-country study designs.”

**

COMMENT

Line 194-195: It might be useful to clarify how species were determined to be invasive – was this based on studies that specifically mentioned invasive species monitoring? If not, was the definition of ‘invasive’ accounted for across studies that may have included the native range of ‘invasive’ species e.g. Aedes aegypti?

RESPONSE

Thank you for this important recommendation. We revised Results section 3.3 to clarify that this classification was based on how the original studies described the target species within their specific geographic and surveillance contexts, and revised the wording accordingly:

Results section 3.3:

“Among the included studies, mosquito monitoring was the most common focus (n=40, 65.6%), primarily concentrated on mapping the distribution, abundance, and spread of mosquito taxa, including species described by the original studies as invasive or nuisance in specific geographic and surveillance contexts. Aedes albopictus was the most commonly reported species (n=16, 26.2%)2,7,10,15,22,24,26–28,35–40,45, followed by Aedes japonicus (n=6, 9.8%)2,23,25,26,38,39, Aedes koreicus (n=5, 8.2%)2,10,32,38,39, and Aedes aegypti (n=3, 4.9%)2,26,47. Additional species included Aedes notoscriptus (n=1, 1.6%)66 and Culex species, most commonly Culex pipiens (n=4, 6.6%)2,10,35,41 and Culex quinquefasciatus (n=1, 1.6%)66.

A smaller subset of studies focused on Anopheles species, including Anopheles plumbeus (n=2, 3.3%)30,31, members of the Anopheles gambiae complex (A. gambiae s.s., A. arabiensis, A. funestus; n=1, 1.6%)60, and mixed Anopheles larvae observations through community surveillance platforms (n=3, 4.9%)11,74,75.”

**

COMMENT

Line 211: Maybe the number of studies from the Americas and Oceania that included surveillance for dengue/ Zika/ Chikungunya could also be provided, for consistency.

RESPONSE

Thank you for this important observation. We revised Results section 3.3 to provide the corresponding counts for studies from the Americas and Oceania that focused on surveillance targeting dengue, Zika, or chikungunya.

Results section 3.3:

“Several studies from the Americas (n=5 of 13, 38.5%) and Oceania (n=2 of 7, 28.6%) also focused on dengue, Zika, or chikungunya surveillance.”

**

COMMENT

Line 240: It would be interesting to know what other traps were used and the breakdown of trap types by number of studies.

RESPONSE

Thank you for this helpful suggestion. We revised Results section 3.5.1 to provide the breakdown of trap types used across the included studies:

Results section 3.5.1:

“Trap-based data collection was also reported in 11 studies (n=11, 18.0%), most commonly using ovitraps (n=5 of 11, 45.5%). Other trap types included BG-Sentinel traps (n=3, 27.3%), BG Gravid Aedes Traps (BG-GAT) (n=3, 27.3%), CO₂-baited traps (n=2, 18.2%), and BG-Mosquitaire and CDC light traps (n=1 each, 9.1%). Some studies used more than one trap type, so the total number of trap occurrences exceeds the number of studies.”

**

COMMENT

Line 271-272: Maybe the last three approaches can be reordered to reflect decreasing frequency (public websites, social media campaigns, institutional newsletters).

RESPONSE

Thank you. We revised the order of the last three recruitment approaches in Results section 3.5.5 to reflect decreasing frequency:

Results section 3.5.5:

“…public websites or email invitations (n=5, 8.2%)26,27,30,35,44, social media campaigns (n=4, 6.6%)10,45,61,63, and institutional newsletters or academic channels (n=3, 4.9%)44,61,63.”

**

COMMENT

General comment: The discussion lists several biases of citizen science data. While not necessary, it could be interesting to give an overview (in the results) of biases that the studies themselves have identified in their work.

RESPONSE

We thank the reviewer for this helpful suggestion. We agree and added a new Results subsection, Section 3.9 (“Reported biases and methodological limitations”), to summarize the main biases and limitations explicitly reported by the included studies. We also revised Methods Sections 2.4 and 2.5, and the Abstract accordingly:

Results section 3.9 (added):

“3.9 Reported biases and methodological limitations

Across the included studies, several explicitly reported biases or methodological limitations affecting their own citizen science data or implementation. Data quality issues were the most common (n=35, 57.4%), followed by validation limitations (n=32, 52.5%), recruitment/engagement challenges (n=32, 52.5%), and participation and sampling biases reported by the studies themselves (n=27, 44.3%), which included uneven observation or reporting effort. Representativeness limitations, indicating that the contributed data may not adequately reflect the full target population or study context, were noted in 25 studies (n=25, 41.0%). Technology-related barriers (n=16, 26.2%) and spatial bias (n=15, 24.6%) were reported less often.”

Methods section 2.4 (revised):

“…v) reported biases, methodological limitations, and ethical considerations (e.g. participation and sampling biases, data quality issues, validation limitations, recruitment/engagement challenges, representativeness limitations, technology-related barriers, spatial bias, ethical procedures, and data availability).”

Methods section 2.5 (revised):

“…reported biases and methodological limitations, and ethical considerations.”

Abstract section (revised):

“…reported biases and methodological limitations, and ethical and governance practices.”

**

############################################################################

Responses to Reviewer #2

Comments to the Author

This manuscript presents a scoping review examining the role of citizen science in mosquito-borne disease surveillance and control. The authors analyze 61 studies published between 2000 and 2025 that use citizen participation to collect mosquito-related data or support vector surveillance initiatives. The review summarizes study characteristics, technologies used, participation strategies, analytical outputs, and governance practices. The topic is timely and highly relevant. Citizen science initiatives have become an increasingly important complement to traditional mosquito surveillance programs, particularly with the widespread availability of mobile technologies and digital reporting platforms. The manuscript provides a useful overview of existing initiatives and highlights the rapid growth of citizen-science projects in this field. However, several aspects that determine the scientific robustness and operational value of citizen-science surveillance systems are insufficiently addressed in the analysis and would benefit from further development.

RESPONSE

We sincerely thank Reviewer #2 for their thoughtful summary and positive evaluation of our work. We are grateful for the constructive feedback which has contributed to strengthening the overall quality and rigor of our study. In the revised manuscript, we have carefully addressed the reviewer’s suggestions and incorporated additional clarifications where appropriate. Major changes have been highlighted for ease of review.

**

COMMENT

A first aspect concerns the discussion of ethical and governance issues. In the current manuscript, governance considerations are mainly addressed in terms of whether ethical procedures are reported by the studies included in the review. While this is important, the discussion could be expanded to consider the role of ethical transparency as a mechanism to foster participation and trust among citizens. Citizen-science initiatives depend fundamentally on voluntary engagement from the public, and participation levels are strongly influenced by the degree of trust participants place in the project. Clear communication about how data are collected, how they will be used, and how personal information is protected is, therefore, not only an ethical requirement but also a practical mechanism to encourage participation. In many mosquito citizen-science platforms, personal data collection is deliberately minimized or avoided, and the explicit communication of this principle can serve as an important incentive for participation. Addressing how different projects communicate these governance and privacy practices to participants would therefore provide a more nuanced perspective on the role of ethics and governance in citizen-science initiatives. Linked to that, as the manuscript highlights, many studies are related to the same citizen science project or use the same platform, and authors should make sure that in all these cases, some basic information related to ethical or data collection issues should not be mentioned, considering that it has been previously described or explained in previous papers. Saying that to make note of the authors to check for consistency for platforms or projects beyond what is explained or described in a single paper. For example, checking Supplementary file 2 “Extraction table” it is possible to see that kind of inconsistencies between study 1 and 2, both use Mosquito Alert tools, in the 1st ethics about the obtention of pictures and reports is stated (e.g. user agreement, anonymous participation, etc.), while in the 2nd, even if also use Mosquito Alert for image classification, the obtention of the images appears as “Not stated”. If they used the Mosquito Alert app to collect images, it is supposed that reports have been collected with user agreement and anonymous participation again. I will recommend the authors in that table to create a clear field identifying the project or tool used to unify the studies and check that information is consistent across all studies for the same platform and tool.

RESPONSE

We thank the reviewer for this thoughtful and important comment. We agree that the previous version focused mainly on whether ethical procedures were explicitly reported by individual studies and did not sufficiently consider ethical transparency as a mechanism for fostering trust and participation. We also agree that recurring platforms may embed governance and privacy features that are not restated in every publication. To address this, we revised the Discussion, expanded Results section 3.10, and updated S2 File by adding a platform-level ethics column and improving consistency across studies using the same platform. The following revisions were made:

Results section 3.10 (added):

“Notably, several recurring digital platforms operated under established governance frameworks that commonly addressed consent, privacy, anonymization, visibility, and data sharing. As these provisions were often embedded within the platform itself, they were not always restated explicitly in every individual study using the same tools. For example, Mosquito Alert incorporated user-agreement and privacy-policy acceptance, anonymous participation, random user identifiers, geolocation protection, and open-data or licensing terms. GLOBE Observer used account-based participation with public screen names (rather than public personal identifiers) and screened submitted photographs for faces or identifying text. Mückenatlas used explicit consent procedures, confidential handling of personal data, anonymized sharing of mosquito and location information, and opt-in public visibility. Similarly, iNaturalist allowed users to retain ownership of submitted content while selecting licenses and controlling geoprivacy settings for observations.”

Discussion section (added):

“Beyond reporting practices, ethical transparency plays a central role in citizen science systems by shaping participant trust, engagement, and sustained contribution. Because these initiatives rely on voluntary public participation, clear communication about what data are collected, how they are used, how privacy is protected, and whether submissions are anonymized or shared openly is not only an ethical requirement but also a practical mechanism for encouraging participation. Many mosquito-focused citizen science platforms deliberately minimize personal data collection, use anonymized or masked identifiers, or allow users to control visibility and sharing settings. When clearly communicated, these protections may therefore strengthen confidence in participation and support sustained data contribution. In this context, transparent and standardized reporting of governance practices would provide a better understanding of how

---

## [Decision Letter · Decision Letter 1]

20 Apr 2026

The Role of Citizen Science in Mosquito-Borne Disease Surveillance and Control: A Scoping Review

PONE-D-26-07716R1

Dear Dr. Mollalo,

We’re pleased to inform you that your manuscript has been judged scientifically suitable for publication and will be formally accepted for publication once it meets all outstanding technical requirements.

Kind regards,

Daniel de Paiva Silva, Ph.D.

Academic Editor

PLOS One

Additional Editor Comments (optional):

Dear Dr. Mollalo,

After this review round, I am pleased to accept your manuscript for publication in PLoS One.

Sincerely,

Daniel Silva

Reviewers' comments:

Reviewer's Responses to Questions

**Comments to the Author**

1. If the authors have adequately addressed your comments raised in a previous round of review and you feel that this manuscript is now acceptable for publication, you may indicate that here to bypass the “Comments to the Author” section, enter your conflict of interest statement in the “Confidential to Editor” section, and submit your "Accept" recommendation.

Reviewer #1: All comments have been addressed

Reviewer #2: All comments have been addressed

2. Is the manuscript technically sound, and do the data support the conclusions?

Reviewer #1: (No Response)

Reviewer #2: Yes

3. Has the statistical analysis been performed appropriately and rigorously? 

Reviewer #1: (No Response)

Reviewer #2: Yes

4. Have the authors made all data underlying the findings in their manuscript fully available?

Reviewer #1: (No Response)

Reviewer #2: Yes

5. Is the manuscript presented in an intelligible fashion and written in standard English?

Reviewer #1: (No Response)

Reviewer #2: Yes

6. Review Comments to the Author

Reviewer #1: The authors have addressed all comments carefully and the manuscript has been improved as a result. The findings are introduced, presented and discussed very clearly and in my opinion the manuscript is worthy of publication.

Reviewer #2: Based on the revised manuscript and the point-by-point response, I consider that the authors have properly addressed my main comments. In particular, the revised version now gives a much stronger and more integrated treatment of the issues I had raised regarding ethical transparency and governance, participation bias, effort-bias handling, validation mechanisms, and the need for a more synthetic framework linking participation, validation, analytical outputs, and the functional roles of citizen-science platforms. These additions are reflected in the expanded abstract and methods, as well as in the new or substantially revised Results sections on validation mechanisms, effort-bias handling, reported biases and methodological limitations, and ethical/governance issues, together with a more integrated discussion. Overall, these changes clearly improve the scientific robustness and interpretability of the review. Reviewer 1’s comments also appear to have been addressed appropriately.

From my point of view, the new version provides a clearer picture of the diversity of citizen-science initiatives and the different surveillance and public-health functions they can serve. Compared with the previous version, the revised manuscript is more complete and analytically stronger, so I do not have any major remaining concerns, and I believe the manuscript is now suitable for acceptance. Congratulations to the authors for this valuable piece of work.

7. PLOS authors have the option to publish the peer review history of their article (what does this mean?). If published, this will include your full peer review and any attached files.

Reviewer #1: No

Reviewer #2: **Yes:** Alex Richter-Boix

---

## [Editor Report · Acceptance letter]

PONE-D-26-07716R1

PLOS One

Dear Dr. Mollalo,

I'm pleased to inform you that your manuscript has been deemed suitable for publication in PLOS One. Congratulations! Your manuscript is now being handed over to our production team.

Kind regards,

on behalf of

Dr. Daniel de Paiva Silva

Academic Editor

PLOS One